1RESEARCH ARTICLE

# A role for the cerebellum in the control of verbal interference: Comparison of bilingual and monolingual adults

**Roberto Filippi** [1,2]*, **Eva Periche Tomas**[1,2], **Andriani Papageorgiou**[1,2], **Peter Bright**[2,3]

**1** Institute of Education, University College London, London, England, United Kingdom, **2** MULTAC (Multilanguage and Cognition Lab), Institute of Education, University College London, London, England, United Kingdom, **3** Anglia Ruskin University, Cambridge, England, United Kingdom

* r.filippi@ucl.ac.uk

**Data Availability Statement:** The data are available on Figshare (DOI: 10.6084/m9.figshare. 12024795).

**Funding:** This work was supported by the Leverhulme Trust UK [RPG-2015-024], obtained

## Abstract

We evaluate brain structure sensitivity to verbal interference in a sentence interpretation task, building on previously reported evidence that those with better control of verbal interference show higher grey matter density in the posterior paravermis of the right cerebellum. We compare brain structure sensitivity to verbal interference control across two groups, English monolingual (N = 41) and multilingual (N = 46) adults. Using voxel-based morphometry, our primary goal was to identify and explore differences in regional patterns of grey matter sensitivity to performance on the sentence interpretation task, controlling for group variability in age, nonverbal reasoning and vocabulary knowledge. There was no group difference in performance but there was a significant group effect in grey matter sensitivity to task performance in our region of interest: stronger sensitivity in the paravermis in bilinguals compared to monolinguals in accuracy performance in the high (relative to low) verbal interference condition. This effect was observed when the linguistic interference was presented in an unfamiliar language (Greek) but not when presented in the familiar language (English). Our findings suggest that multilanguage acquisition mediates regional involvement within the language network, conferring enhanced functional plasticity within structures (including the paravermis) in the service of control of linguistic interference.

## Introduction

Auditory control of interference is the cognitive ability to inhibit the processing of concurrent irrelevant auditory stimuli. This ability could be particularly beneficial for people who have to monitor, process and communicate in more than one language (i.e., the bilingual population). The aim of the current study was to use MRI structural imaging to investigate the neural correlates of verbal interference during speech comprehension in monolingual and bilingual populations, with specific focus on the cerebellum. Previous studies looking into the control of verbal interference have highlighted the involvement of the left caudate during an inhibitory control task based on colour naming [1] and left inferior or middle frontal regions when bilinguals were making decisions on a semantic written task in their native language [2,3]. However, until recently, the literature has been relatively silent on the candidate role of the

by RF and PB. https://www.leverhulme.ac.uk/ The funders had no role in the design or any other components of this study.

**Competing interests:** The authors have declared that no competing interests exist.

cerebellum in this function, perhaps due in part to the tendency in many functional MRI studies to remove the cerebellum in its entirety prior to statistical analysis. Over the past decade or so functional, lesion and developmental studies have shown and highlighted the importance of this structure in language processing [4,5,6,7,8].

A specific area in the cerebellum, the most medial part of lobule VIIIA that lies within the posterior paravermis has been associated with the control of motor movements [9]. Furthermore, a more recent study has shown that the right posterior paravermis is involved in the control of verbal interference [10] indicating a role for this structure in both motor and language functions. These authors employed structural neuroimaging in a sample of Italian bilingual participants (mean age 33 years old) who were late English learners, to identify brain regions that were positively correlated with the ability to control verbal interference. This was achieved through the administration of a dichotic listening task primarily adapted from a paradigm designed by Bates and colleagues [11,12] in which the participants were required to identify the agent in a series of sentences that differed in terms of structural complexity and in the absence and/or presence of interfering sentences. Results indicated that a cerebellar area, the right posterior paravermis, had higher grey matter density in those who were better at controlling verbal interference. These findings were consistent with a functional imaging study which investigated areas of activation when bilingual German participants (age range 23–62), who had been speaking English for more than 4 years, completed a semantic decision task while reading a list of words in the presence of distracting written stimuli [10].

Filippi and colleagues' work provided novel data about structural and functional effects on a specific cerebellar area in the control of verbal interference. However, the sample used in that study was a group of highly-proficient bilingual adults. To date, the question of whether these effects are a unique characteristic of the bilingual population or can also be observed in monolinguals has remained unanswered. Therefore, the rationale to conduct the current study is to address this question by comparing the performance of English monolingual and bilingual adults. This approach allows us to systematically address whether multilanguage acquisition causes structural change observed within the cerebellum.

The procedure implemented in the Filippi et al. [10] study was followed, building on the experimental tasks and the target population. In this new study, the sample included a large group of English monolingual speakers who were compared with bilinguals from different linguistic backgrounds. Any potential differences within the two language groups were explored, both at behavioural and neural levels. This study further modified the original experimental design by increasing the upper age threshold from approximately 40 to 80 years old in order to incorporate possible age-related structural differences in the region of interest (ROI).

## Methods

### Participants

This project has been approved by the Science and Technology Research Ethics panel at Anglia Ruskin University (FST/FREP/15/505). The study included 87 right-handed adults between 18 and 80 years old (48 females, mean age 45.9 SD = 19.3) of which 41 were English monolinguals and 46 were bilinguals from different linguistic backgrounds. All participants were residents in the UK at the time of testing. Bilingual participants completed a language history questionnaire adapted from Papageorgiou et al. [13], which revealed that, overall, the group had high levels of English language proficiency (see Table 1).

The study was approved by the local ethics committee and all participants gave written informed consent.

**Table 1. Bilingual participants' linguistic information.**

| Linguistic background | First language | Catalan ($n = 1$) |
| --- | --- | --- |
| | | Danish ($n = 1$) |
| | | Dutch ($n = 2$) |
| | | English ($n = 21$) |
| | | French ($n = 4$) |
| | | German ($n = 3$) |
| | | Hebrew ($n = 1$) |
| | | Italian ($n = 4$) |
| | | Mandarin ($n = 2$) |
| | | Polish ($n = 2$) |
| | | Portuguese ($n = 1$) |
| | | Russian ($n = 2$) |
| | | Spanish ($n = 2$) |
| | Second language | Arabic ($n = 1$) |
| | | English ($n = 25$) |
| | | Finnish ($n = 1$) |
| | | French ($n = 9$) |
| | | German ($n = 2$) |
| | | Hungarian ($n = 1$) |
| | | Italian ($n = 2$) |
| | | Polish ($n = 1$) |
| | | Spanish ($n = 3$) |
| | | Urdu ($n = 1$) |
| | Age of first exposure | Birth-6 years ($n = 27$) |
| | | 7–12 years ($n = 19$) |
| | Average Years of learning/Using L2 | 27 |
| Self-rated proficiency (1–7) | Reading | $M = 6.1$; $SD = 0.7$ |
| | Writing | $M = 5.6$; $SD = 0.9$ |
| | Speaking | $M = 5.9$; $SD = 0.8$ |
| | Listening | $M = 6.2$; $SD = 0.7$ |

## Materials and procedure

Participants completed a background online questionnaire [14,15] and were pre-screened for MRI safety before attending the testing sessions. Eligible participants were invited to complete a behavioural and structural imaging session which took place at UCL. The behavioural procedure included a battery of 4 tasks that measured vocabulary knowledge in English language (BPVS III), non-verbal reasoning (Raven's), verbal working memory (Digit span) and an auditory interference task (Sentence interpretation task). All information and instructions were given in English.

## BPVS III (British picture vocabulary scale III)

This task uses 14 sets of 12 slides [16]. Each slide presents four pictures (1 target and 3 distracters) with an auditory cue corresponding to the target image. All participants started from set 10, designated for ages of 14 and above and progressed to the next set only if all items where correct. One or more errors in set 10 results in moving to set 9 and this rule applies until all items within a set are correct. The task was completed if they reached set 14 or if they made 8

mistakes in a particular set. The final score was computed by subtracting the number of errors from the highest possible score.

### Raven's advanced progressive matrices task (set 1)

This test of non-verbal fluid intelligence includes 12 increasingly complex trials [17] with an estimated completion time of ten minutes. Each trial involves identifying the missing part from a geometric shape. Participants were required to select the correct part from eight candidate choices. One point is awarded for each correct answer.

### Digit span

Participants are instructed to listen to a sequence of numerical digits [18]. The aim is to repeat the sequence verbatim or in reverse order as instructed by the researcher. Participants were presented with two sequences of two digits. If at least one sequence was answered correctly the next set of sequences including one additional digit was subsequently presented. This was repeated until either two consecutive mistakes for a given sequence length were made or a sequence with nine digits was reached. This process was followed first for the forwards and then for the backwards condition. Scores were recorded according to the number of sequences answered correctly.

### Sentence interpretation task

The sentence interpretation task [14,15] was programmed and conducted using E-Prime (version 2.0; [19]). This study extended Filippi et al.'s [14,15] sentence interpretation task to the comprehension of English target sentences in the presence of competing speech in English or in Greek, a language that was not known by any participant. Each sentence featured two animals and participants were asked to determine which of the two animals was performing a bad action towards the other (e.g., biting, bumping, pushing). For example, in the sentence *the rabbit is scared by the horse*, the horse is the bad animal. The animals featured in the target sentence simultaneously appeared on the screen, one animal on the left and one on the right; the participant pressed the corresponding left/right button on the gamepad to indicate which animal was *being bad* (see Fig 1).

The target sentences varied in their syntactic complexity, which altered their cognitive load [20]. The *easy* canonical sentences (Subject-Verb-Object: S-V-O) are less cognitively demanding than the *hard* non-canonical sentences (Object-Verb-Subject: O-V-S or Object-Subject-Verb: O-S-V; [20]). Examples sentences are shown in Table 2.

Each of the English target sentences were presented binaurally with one of three levels of audible interference: (1) no interference–control condition, (2) English speaking interference, (3) Greek speaking interference. Competing sentences in English and Greek always featured a different animal and had different grammatical constructions. A pseudorandom pairing of target and interference sentences ensured that there was no overlap within trials between target animals and actions and interfering animals and actions.

Participants were provided with instructions that included listening to the sentence first and then choosing the bad animal. A practice run was first presented which included two blocks, one with a male and the other one with a female voice, uttering the target sentence. Participants were instructed to listen only to the voice stated at the beginning of each block and ignore the voice of the opposite gender.

Block presentation for the two practice blocks and two experimental blocks were counterbalanced so that half of the participants had a male target voice for the first block and female for the second block and vice versa). The practice blocks included eight trials each and the

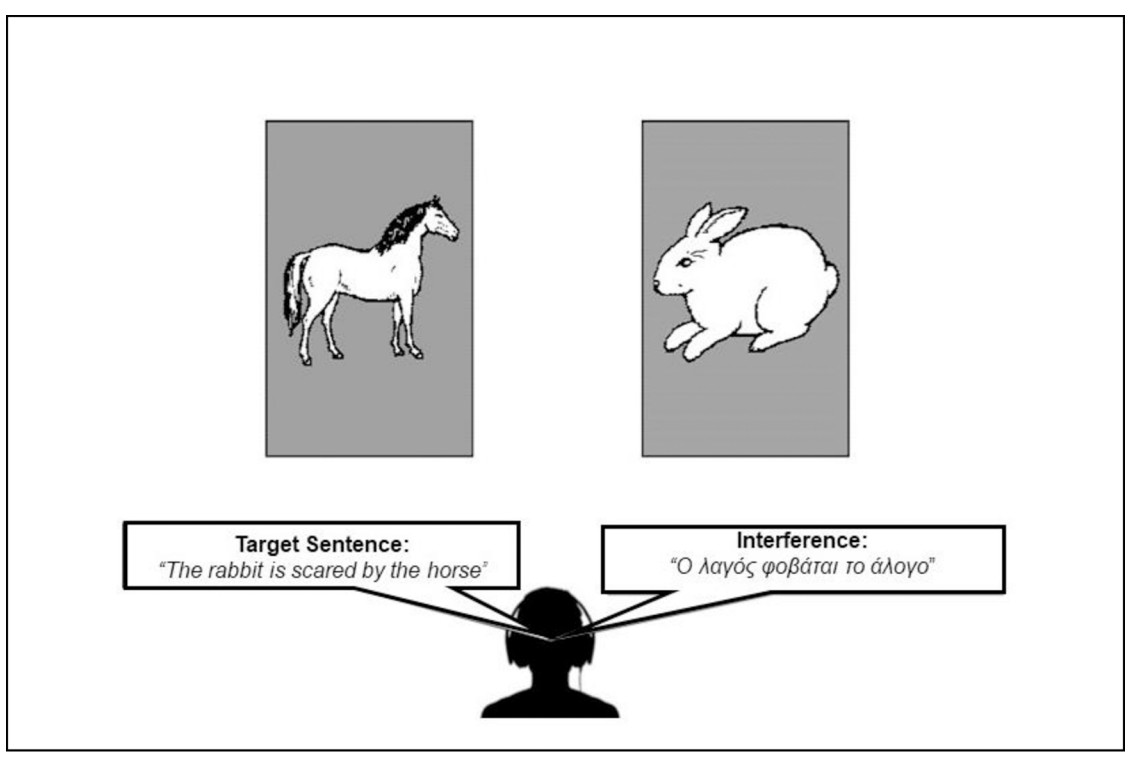

**Fig 1. An illustration of the sentence interpretation task.** Participants were instructed to indicate the animal doing the bad action as fast and accurately as possible. Target sentences were always in English. Competing sentences were either in English or in Greek and always spoken by the opposite gender (e.g., target: man's voice, interference: woman's voice or vice-versa). Control trials were administered without verbal interference.

experimental block included 144 trials. For each trial the animal stimuli appeared on the screen for a maximum of 3000 ms or until a response was selected.

Visual stimuli consisted of black and white animal drawings on a grey background (70 mm x 50 mm) taken from picture databases [21,22]. Verbal auditory stimuli were 144 target sentences spoken by an English female, 144 target sentences spoken by an English male, 24 English interference sentences spoken by an English female and 24 English interference sentences spoken by an English male, 24 Greek interference sentences spoken by a Greek female and 24 Greek interference sentences spoken by a Greek male. Trials were presented in a random order.

## Structural image acquisition

Whole brain anatomical T1-weighted images were acquired on a Siemens Sonata scanner housed in the Birkbeck-UCL Centre for Neuroimaging (BUCNI). For each of the 87 participants, 176 sagittal slices were collected with a 256 x 224mm image matrix, providing a 1mm$^3$ voxel resolution (TR/TE/TI = 12.24/3.56/530ms).

**Table 2. Example of sentence type (the agent is in bold).**

| Sentence Type | Constituent Order | Sentence |
|---|---|---|
| **Canonical** | Active (S-V-O) | The **whale** is pushing the frog |
| | Subject Cleft (S-V-O) | It's the **seal** that is pushing the cow |
| **Non-canonical** | Passive (O-V-S) | The whale is pushed by the **seal** |
| | Object cleft (O-V-S) | It's the monkey that the **cow** is pushing |

## Structural image analysis

Pre-processing was applied using the Computational Anatomy Toolbox (cat12) for SPM12 (http://www.neuro.uni-jena.de/cat/). After manual inspection to ensure approximate alignment, initial segmentation was applied using the SPM tissue probability maps and subsequently the tissue probability atlases provided by the International Consortium for Brain Mapping (ICBM). Spatial registration was undertaken in MNI space using a DARTEL template derived from 555 healthy control subjects in the IXI-database (https://brain-development.org/ixi-dataset/) and the shooting template supplied as part of the cat12 toolbox. In order to correct for volumetric changes associated with nonlinear spatial normalisation, voxel values in the segmented grey matter were multiplied by the Jacobian determinant (volume changes) derived from the spatial normalisation stage. These modulated images, therefore, retained the total amount of grey matter from the original images, thereby providing the basis for analysis of volumetric differences between monolingual and bilingual brains and detection of regional volume sensitivity to behavioural performance on our tasks. Images were normalised to Montreal Neurological Institute (MNI) stereotaxic space, and smoothed using an isotropic kernel of 8 mm at full-width half-maximum (FWHM).

## Statistical analysis of structural data

A full factorial interaction model was built to identify group effects (ML vs BL) while factoring out variance associated with i. age, ii. Raven's Matrices raw score, iii. BPVS III scores. Mean task scores on the non-canonical (i.e., 'difficult'), canonical ('easy') interference and control trials in the sentence interpretation task were entered as covariates separately modelled for each group (ML, BL). Scores in both English interference and Greek interference conditions were included in the model. We used volumes of interest within the cerebellum based on findings reported in Filippi et al. [10], as described below.

# Results

## Behavioural analyses

Accuracy scores in the control condition (i.e., without language interference) were subtracted from those in the verbal interference conditions to obtain a task ability score. The scores are on a negative scale because performance on non-interference (baseline) tasks was generally better than that of interference conditions. As a result, a less negative score indicates better ability to manage interference. All analyses were controlled for age, English vocabulary knowledge, working memory and non-verbal reasoning ability scores. For control of interference, participants had worse performance when the task involved non-canonical sentences with English interference. Ability scores (means and standard errors) are reported and illustrated in Fig 2.

A 2\*2\*2 mixed ANOVA with the between-subject factor being *Language group* (bilinguals, monolinguals) and the within-subjects factors being *Sentence type* (canonical, non-canonical) and *Interference type* (*English*, *Greek*) was conducted. ANOVA revealed a main effect of *Interference type*, $F(1,85) = 72.12$, $p < 0.001$, $\eta_p^2 = .46$, and a main effect of *Sentence type*, $F(1,85) = 13.52$, $p = 0.001$, $\eta_p^2 = .14$. However, there was no significant main effect of *Language group*, $F(1,85) = .91$, $p = 0.34$, $\eta_p^2 = .01$, indicating that the English monolingual and the bilingual participants had comparable performance in all experimental conditions.

There were no significant *Language group* \* *Sentence type* and *Language group* \* *Interference type* interactions, $F(1,85) = 2.85$, $p = 0.10$, $\eta_p^2 = .03$; $F(1,85) = .68$, $p = 0.78$, $\eta_p^2 = .01$, respectively. The three-way interaction *Sentence type* \* *Interference type* \* *Language group* was also non-significant, $F(1,85) .71$, $p = 0.40- =$ , $\eta_p^2 = .01$.

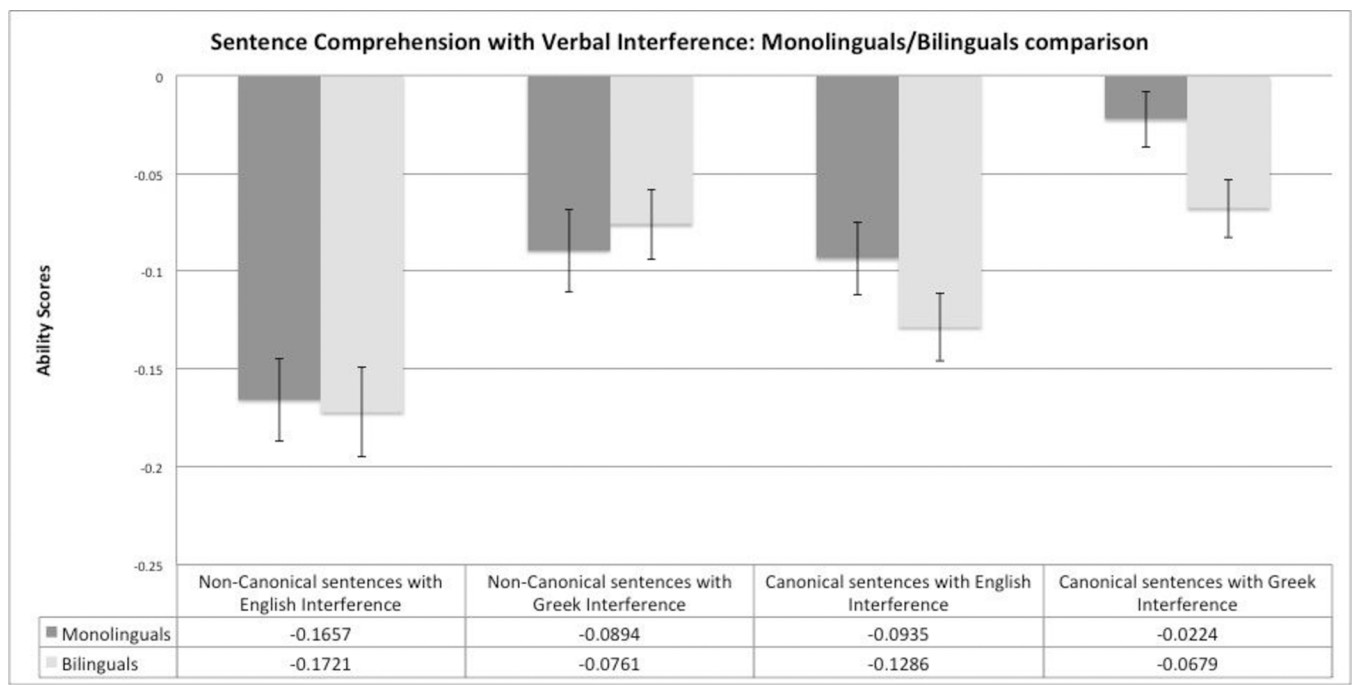

**Fig 2. Monolinguals' and bilinguals' ability scores for errors with SE bars in the sentence interpretation task for both canonical and non-canonical sentences in the presence of English and Greek interference.**

In summary, behavioural analysis for errors did not reveal any significant group differences in comprehension of English target sentences in the presence of English or Greek interference.

## Voxel-based morphometry results

We identified *a priori* volumes of interest within the cerebellum. We took the peak coordinate reported in Filippi et al. [10] where there was a positive correlation between grey matter density and control of interference in the sentence interpretation task, and applied a 10mm radius volume of interest at that coordinate (x = +12 y = -64 z = -42). For completeness we ran analyses in both hemispheres. All reported effects were significant at p = .001, FWE corrected for multiple comparisons at p = .05.

A direct group comparison revealed more grey matter volume in bilinguals over monolinguals within our volume of interest in the right hemisphere as shown in Table 3 and Fig 3. No effects were observed in the left hemisphere, or for monolinguals over bilinguals.

**Volumetric sensitivity to performance on the sentence interference task.** There was no main effect for interference (averaged across English and Greek conditions) relative to control trials or sentence type (canonical vs noncanonical) within our volumes of interest. Furthermore, there was no evidence for differences in volumetric sensitivity to the control of English

**Table 3. Differences in grey matter volume for bilinguals relative to monolinguals.**

| Effect | Region | Peak | Z | P (FWE corr) |
|---|---|---|---|---|
| **Bilinguals > monolinguals** | **Right hemisphere:** | | | |
| | *10mm VOI at x +12 y -64 z -42* | 18–63–48 | 3.5 | .009 |
| | **Left hemisphere:** | | | |
| | *10mm VOI at x -12 y -64 z -42* | - | - | - |

No effects were observed for monolinguals over bilinguals.

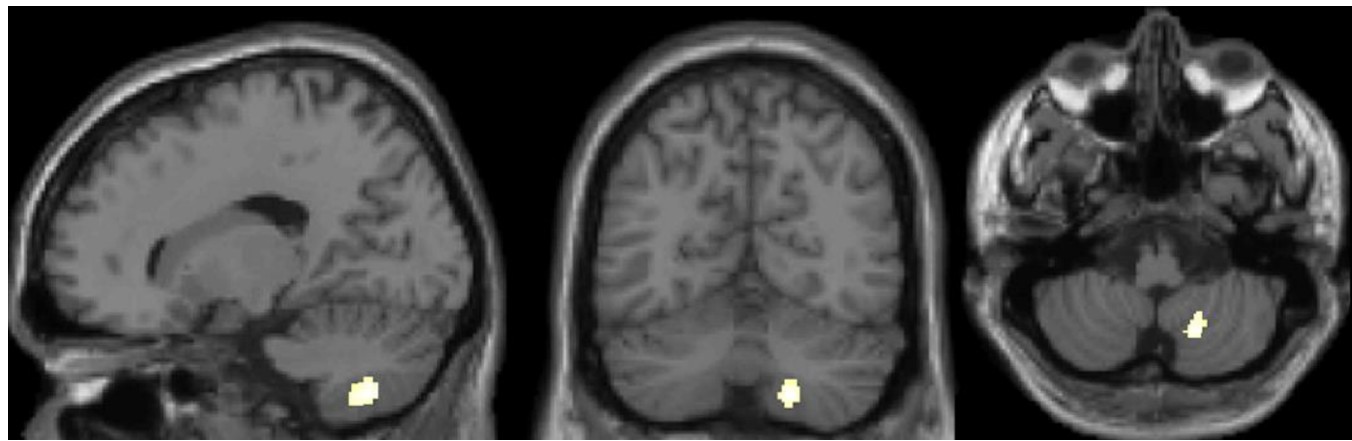

**Fig 3. Group comparison of cerebellar grey matter volume, greater in bilinguals.** Sections taken at x = 18 y = -63 z = -48. There were no effects for monolinguals > bilinguals.

(i.e., familiar) interference relative to control performance within our full sample, within either group or by sentence type (canonical/noncanonical). There was no main effect for control of Greek interference across all participants. However, when we modelled monolingual and bilingual performance on noncanonical sentences relative to control performance separately we found significant volumetric sensitivity in our bilinguals but not our monolinguals and there was a significant group (bilingual/monolingual) x condition (Greek interference/control) interaction. These effects held in both hemispheres, as shown in Table 4. Fig 4C shows the spatial extent of the volumetric sensitivity for the interaction effect below the structural effects (4a) and functional effects (4b) reported in Filippi et al. [10].

The source of the group x condition interaction, in both the right and left VOI is demonstrated in Fig 5. Here, raw beta values (i.e., unadjusted for the other covariates in our model) are plotted against ability scores (interference minus control performance) on the noncanonical sentences. There was a significant positive correlation between ability score and grey matter volume in our bilingual participants, but this correlation was nonsignificant (and negative) in

**Table 4. Volumetric sensitivity within the cerebellum to performance (ability scores) on the sentence interpretation task.**

| Effect | Region | Peak | Z | P (FWE corr) |
|---|---|---|---|---|
| | **Right hemisphere:** | | | |
| **Condition:** | *10mm VOI at x +12 y -64 z -42* | | | |
| **Greek interference vs control:** | | - | - | - |
| **Group: Bilinguals** | | 9–62–51 | 3.40 | .012 |
| **Monolinguals** | | - | - | - |
| **Group x condition interaction** | | 9–63–51 | 3.87 | .003 |
| | **Left hemisphere:** | | | |
| **Condition:** | *10mm VOI at x -12 y -64 z -42* | | | |
| **Greek interference vs control:** | | - | - | - |
| **Group: Bilinguals** | | -15–65–48 | 3.47 | .010 |
| **Monolinguals** | | - | - | - |
| **Group x condition interaction** | | -14–65–50 | 4.74 | < .001 |

There were no group differences in grey matter sensitivity to ability scores in the context of English interference and no areas of the cerebellum that showed greater sensitivity to task performance in monolinguals relative to bilinguals.

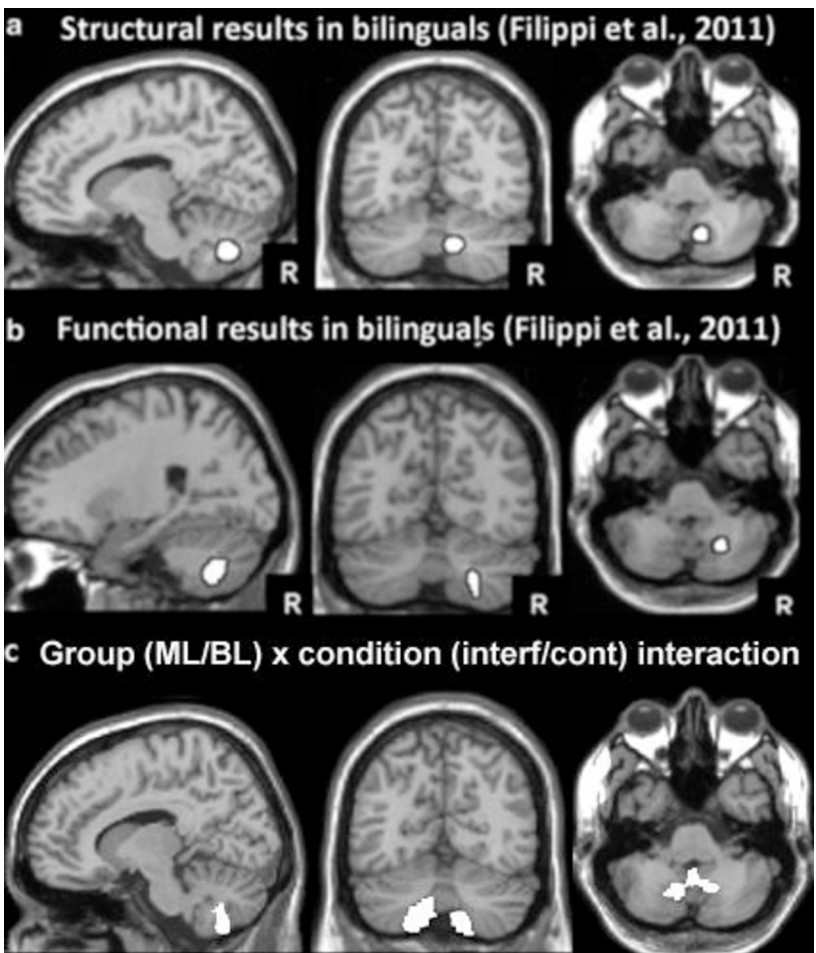

**Fig 4.** a) Structural imaging results from Filippi et al. [10] shown at x = 12, y = -64, z = -42. White area shows where grey matter density was higher in those bilinguals (N = 26) with better control of interference; b) Functional imaging results from Filippi et al. [10] shown at x = 20, y = -64, z = -42. White area shows where activation was higher when German participants (N = 8) performed semantic decisions on written words in native vs English interference; c. Results from the current study shown at x = 12, y = -64, z = -42. White area shows where the difference in volumetric sensitivity across conditions (Greek interference and control trials) differed significantly as a function of group (bilingual/monolingual).

our monolinguals, a pattern which held true in both hemispheres. In both VOIs, using the Fisher r-to-z transformation, correlation coefficients were significantly stronger in the bilingual sample (LH: z = 3.19, p < .01; RH: z = 2.78, p < .01, two-tailed).

## Discussion

In this study, we investigated the role of cerebellar areas in the control of interference in speech comprehension. We extended our previous research (which employed only a bilingual sample) to include a sample of English monolinguals. This new design allowed us to systematically address whether multilanguage acquisition causes structural change observed within the cerebellum.

We acquired raw structural images from a large group of adult individuals (N = 87), split in two groups: English monolinguals (N = 41) and bilinguals of different linguistic backgrounds (N = 46). Their age ranged from 18 to 80 years old. Voxel-based morphometry was used to identify grey matter markers of processing ability on brain structure across the adult lifespan. Consistent with previous work [10] we first acquired data on a behavioural test and then

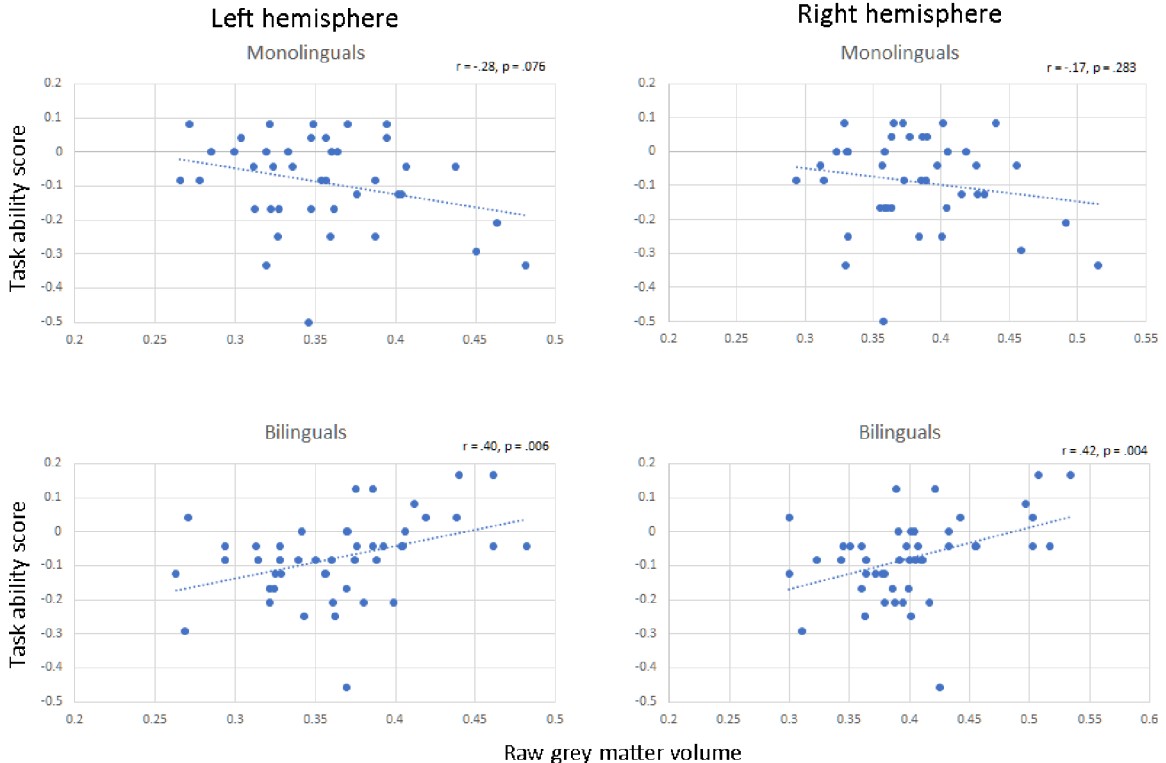

**Fig 5. Plots show correlations of raw beta values (grey matter volume) within 10mm radius spherical ROIs (centred around -12–64–42 and +12–64–42) against task ability score in the context of Greek interference.**

acquired structural MRI data. As a result, all participants performed a sentence interpretation task with different levels of grammatical complexity (canonical and non-canonical sentences), in the presence of verbal interference outside the scanner. Verbal interference was generated by the simultaneous presentation of competing speech in English and Greek. Statistical analysis on behavioural data, that is, ability scores in sentence comprehension accuracy, did not provide any significant difference between the two groups in all experimental conditions. MRI results demonstrated greater absolute volume and volumetric GM sensitivity to Greek interference in bilinguals relative to monolinguals in the same area of lobule VIII reported by Filippi et al. [10].

## In what ways does this study differ from our previous findings?

Our findings support the claim [10] that the cerebellum is functionally involved in controlling verbal interference. Unlike the earlier study involving only bilinguals, however, the task sensitivity effects were bilateral, perhaps indicating shared/interhemispheric functionality rather than strong lateralisation for language processing and control in the cerebellum. Moreover, in the current study the structure-performance correlations were observed in the modulated rather than unmodulated data, indicating volumetric rather than density sensitivity in the cerebellum.

Our data support the claim that the acquisition of a second language impacts on the distributed networks underpinning language processing but also that these effects are not associated with better performance, as indicated by statistically equivalent performance across the groups. Thus, while we have detected robust differences in volumetric sensitivity within the cerebellum in the control of verbal interference, as well as greater absolute cerebellar volume in bilinguals, there

was no cognitive advantage at the behavioural level. These findings are inconsistent with our earlier behavioural work with the sentence interference task [14]. However, in the present study, the task was modified to include a linguistic but semantically non-meaningful condition (i.e., in which the interference was presented in Greek, a language unfamiliar to all participants). Moreover, rather than testing a relatively homogenous group of bilinguals (Italian/English in the 2012 study) and presenting target sentences in the first language in the bilingual group (Italian) and English in the monolingual group, the bilinguals in the present study were drawn from a broad range of linguistic groups and the target sentences were presented only in English. Given these systematic differences in task design and sample characteristics, a clear-cut interpretation of the inconsistent behavioural results across our studies is not straightforward. It is possible that the simpler task design and more constrained sample characteristics employed in our 2012 study may have conferred greater sensitivity for detecting group effects but this is purely speculative. Nevertheless, our current set of findings are consistent with other authors who have also highlighted a complex and ambiguous relationship between structural variability and performance/ability [23,24,25,26,27,28]. These authors have provided evidence that contradicts earlier findings for a possible bilingual advantage in cognitive control (see [29], for an extensive review).

## Why did we find the effect in the control of unfamiliar (Greek) linguistic interference?

We expected that structure/performance correlations would be strongest when controlling interference in the familiar target language (English sentences with English interference). Contrary to our predictions the effect was reliable only in the unfamiliar language interference condition (English sentences with Greek interference). This unexpected finding suggests that the role of the cerebellum might relate more to the orienting of attention than interference control *per se*. Although speculative, we suggest that bilinguals may show more sensitivity to unfamiliar languages and therefore develop resources to manage the additional processing demands associated with selectively attending to a known target language against unfamiliar linguistic noise. Therefore, the bilateral volumetric sensitivity in the cerebellum may reflect this requirement to selectively attend to the target language while monitoring concurrent unfamiliar verbal information. To our knowledge, whether attention in bilinguals is more likely to be automatically engaged by the linguistic properties of an unfamiliar language is currently unknown but, if so, it might provide a parsimonious explanation for our findings. Nevertheless, we are also alert to the possibility of false positives recently highlighted as particularly problematic in structural MRI studies on bilingualism [30]. Our sample size of 87 (46 bilinguals) is notably larger than most in this literature, and the findings themselves part-replicate our earlier work [10], yet we would encourage replication and attempts to further delineate the role of the cerebellum in bilingual cognition.

## Summary

Traditionally, the cerebellum has not been included in models of language processing. More recent research indicates an important role in the cerebellum in language comprehension [31], language perception [32] and language production [33]. Our paper contributes to this literature, demonstrating that bilingualism confers structural differences within lobule VIII and may play an important role in the resolution of linguistic competition in noisy environments. Further work should explore the role of the cerebellum in selective attention in language processing at a functional level. Research should also exploit structural MRI to resolve the comparative significance of volumetric and density sensitivity within the cerebellum in the control of linguistic interference.

## Acknowledgments

We thank Prof. Cathy Price for her valuable comments. A special thought goes to Prof. Annette Karmiloff-Smith who inspired our research.

## Author Contributions

**Conceptualization:** Roberto Filippi, Peter Bright.

**Data curation:** Roberto Filippi, Eva Periche Tomas, Andriani Papageorgiou, Peter Bright.

**Formal analysis:** Roberto Filippi, Peter Bright.

**Funding acquisition:** Roberto Filippi, Peter Bright.

**Investigation:** Roberto Filippi, Peter Bright.

**Methodology:** Roberto Filippi, Peter Bright.

**Project administration:** Roberto Filippi, Peter Bright.

**Resources:** Roberto Filippi.

**Supervision:** Roberto Filippi, Peter Bright.

**Writing – original draft:** Roberto Filippi, Eva Periche Tomas, Andriani Papageorgiou, Peter Bright.

**Writing – review & editing:** Roberto Filippi, Peter Bright.

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
