## [Decision Letter · Decision Letter 0]

28 Jan 2020

PONE-D-19-34005

A role for the cerebellum in the control of verbal interference: comparison of bilingual and monolingual adults

PLOS ONE

Dear Dr Filippi,

Thank you for submitting your manuscript to PLOS ONE. After careful consideration, we feel that it has merit but does not fully meet PLOS ONE’s publication criteria as it currently stands. Therefore, we invite you to submit a revised version of the manuscript that addresses the points raised during the review process.

As you will see, both reviewers have some concerns: Reviewer 1 (Kenneth R. Paap) suggested minor revisions; Reviewer 2 suggested major revisions. I have read the manuscript myself, and I agree with the reviewers as there are some issues that need to be addressed properly before publication – I refer to the reviews for details. If you will decide to revise your manuscript, please pay particular attention to point 1) of Reviewer 2.

We would appreciate receiving your revised manuscript by Mar 13 2020 11:59PM. To enhance the reproducibility of your results, we recommend that if applicable you deposit your laboratory protocols in protocols.io, where a protocol can be assigned its own identifier (DOI) such that it can be cited independently in the future. For instructions see: http://journals.plos.org/plosone/s/submission-guidelines#loc-laboratory-protocols

We look forward to receiving your revised manuscript.

Kind regards,

Claudio Mulatti, Ph.D.

Academic Editor

PLOS ONE

4. We note you have included tables to which you do not refer in the text of your manuscript. Please ensure that you refer to Tables 3 and 4 in your text; if accepted, production will need this reference to link the reader to the Tables.

Reviewers' comments:

Reviewer's Responses to Questions

**Comments to the Author**

1. Is the manuscript technically sound, and do the data support the conclusions?

Reviewer #1: Yes

Reviewer #2: Yes

2. Has the statistical analysis been performed appropriately and rigorously? 

Reviewer #1: Yes

Reviewer #2: Yes

3. Have the authors made all data underlying the findings in their manuscript fully available?

Reviewer #1: Yes

Reviewer #2: Yes

4. Is the manuscript presented in an intelligible fashion and written in standard English?

Reviewer #1: Yes

Reviewer #2: Yes

5. Review Comments to the Author

Reviewer #1: One purpose of the study was to compare bilinguals to monolinguals in ability to comprehend spoken sentences in the presence of verbal interference. No differences were observed. The authors could say more regarding the implications of these null results. An additional purpose was to examine grey matter density in the cerebellum and how individual differences in density relate to differences in verbal interference. The interesting result with respect to this second question was that there was a significant positive correlation, but only for the bilinguals and only when the interference was from an unfamiliar language (Greek). I did not understand the logic underlying the conjecture that this pattern reflected selective attention rather than the control of interference. The study should be of interest to cognitive neuroscience types who have keen interests in localizing cognitive functions. The fact that the behavioral results showed no difference between bilinguals and monolinguals is surprising.

Reviewer #2: The author's present a work that links structural differences in the vermis is related to control of interference in a sentence interpretation task. Overall, the manuscript finds a link between verbal interference and brain structure. I had three particular comments on the work:

1) It was not clear whether the first author was simply referring to previous work of his when referring to the sentence intepretation task. The original task developed by Bates and Macwhinney was tested in multiple languages and also tested with bilinguals. It seems as if the authors are trying to claim credit for work developed by others in the field.

2) The presence of two groups of over 40 subjects per group is relatively high for fMRI studies. However, recent work in the field particularly from Munson and Hernandez (BRLN, 2019) suggests that even this number of subjects may lead to unreliability of the findings. Specifically, it opens up the possibility of a false positive. This should be discussed at the very least as a limitation. Possible steps for ameliorating the situation could be taken, including using multivariate analyses.

6. PLOS authors have the option to publish the peer review history of their article (what does this mean?). If published, this will include your full peer review and any attached files.

Reviewer #1: Yes: Kenneth R. Paap

Reviewer #2: No

---

## [Author Response · Author response to Decision Letter 0]

4 Feb 2020

We thank both Reviewers for confirming that our work is technically sound, has appropriate control and sample sizes and meets all other requirements for being recognised as a rigorous study.

We address all their specific comments below.

Reviewers' comments:

Reviewer 1: One purpose of the study was to compare bilinguals to monolinguals in ability to comprehend spoken sentences in the presence of verbal interference. No differences were observed. The authors could say more regarding the implications of these null results. 

Our response: We have included further considerations with respect of our previous study (Filippi et al., 2012). The amended text has been added to the manuscript on page 19.

These findings are inconsistent with our earlier behavioural work with the sentence interference task (Filippi et al., 2012). However, in the present study, the task was modified to include a linguistic but semantically non-meaningful condition (i.e., in which the interference was presented in Greek, a language unfamiliar to all participants). Moreover, rather than testing a relatively homogenous group of bilinguals (Italian/English in the 2012 study) and presenting target sentences in the first language in the bilingual group (Italian) and English in the monolingual group, the bilinguals in the present study were drawn from a broad range of linguistic groups and the target sentences were presented only in English. Given these systematic differences in task design and sample characteristics, a clear-cut interpretation of the inconsistent behavioural results across our studies is not straightforward. It is possible that the simpler task design and more constrained sample characteristics employed in our 2012 study may have conferred greater sensitivity for detecting group effects but this is purely speculative. Nevertheless, our current set of findings are consistent with other authors who have also highlighted a complex and ambiguous relationship between structural variability and performance/ability (Paap & Greenberg, 2013; Paap, Johnson and Sawi, 2015; Dunabeitia et al., 2014, Dunabeitia & Carreiras, 2015; Hiltchey & Klein, 2011; Klein, 2015).

-----

An additional purpose was to examine grey matter density in the cerebellum and how individual differences in density relate to differences in verbal interference. The interesting result with respect to this second question was that there was a significant positive correlation, but only for the bilinguals and only when the interference was from an unfamiliar language (Greek). I did not understand the logic underlying the conjecture that this pattern reflected selective attention rather than the control of interference. The study should be of interest to cognitive neuroscience types who have keen interests in localizing cognitive functions. The fact that the behavioral results showed no difference between bilinguals and monolinguals is surprising.

Our response: We have included additional clarification suggesting that attention in bilinguals is more likely to be automatically engaged by unfamiliar linguistic information and made this point on page 20 (amended text below)

To our knowledge, whether attention in bilinguals is more likely to be automatically engaged by the linguistic properties of an unfamiliar language is currently unknown but, if so, it might provide a parsimonious explanation for our findings. 

-----

Reviewer 2: The author's present a work that links structural differences in the vermis is related to control of interference in a sentence interpretation task. Overall, the manuscript finds a link between verbal interference and brain structure. I had three particular comments on the work:

It was not clear whether the first author was simply referring to previous work of his when referring to the sentence intepretation task. The original task developed by Bates and Macwhinney was tested in multiple languages and also tested with bilinguals. It seems as if the authors are trying to claim credit for work developed by others in the field.

Our response: This was certainly not our intention and we apologize for giving this impression. In our previous work we provide ample reference to the work of Bates and others. We have now included reference to the paradigm developed by Bates and colleagues which underpins our task. Please find the amended text below and on page 4 in the manuscript.

This was achieved through the administration of a dichotic listening task primarily adapted from a paradigm designed by Bates and colleagues (Bates et al., 2001; MacWhinney & Bates, 1989) in which the participants were required to identify the agent in a series of sentences that differed in terms of structural complexity and in the absence and/or presence of interfering sentences.

The presence of two groups of over 40 subjects per group is relatively high for fMRI studies. However, recent work in the field particularly from Munson and Hernandez (BRLN, 2019) suggests that even this number of subjects may lead to unreliability of the findings. Specifically, it opens up the possibility of a false positive. This should be discussed at the very least as a limitation. Possible steps for ameliorating the situation could be taken, including using multivariate analyses.

Our response: We thank the Reviewer for alerting us to this important work by Munson & Hernandez (2019). We have included consideration in the amended text below (page 20 in the manuscript).

Nevertheless, we are also alert to the possibility of false positives recently highlighted as particularly problematic in structural MRI studies on bilingualism (Munson & Hernandez, 2019). Our sample size of 87 (46 bilinguals) is notably larger than most in this literature, and the findings themselves part-replicate our earlier work (Filippi et al., 2011), yet we would encourage replication and attempts to further delineate the role of the cerebellum in bilingual cognition.

We considered our manuscript improved and thank the Reviewers for their very useful comments. We hope that this revised manuscript is now considered suitable for publication in PLoS ONE. 

Yours sincerely,

Roberto Filippi & Peter Bright

---

## [Decision Letter · Decision Letter 1]

20 Mar 2020

A role for the cerebellum in the control of verbal interference: comparison of bilingual and monolingual adults

PONE-D-19-34005R1

Dear Dr. Filippi,

We are pleased to inform you that your manuscript has been judged scientifically suitable for publication and will be formally accepted for publication once it complies with all outstanding technical requirements.

With kind regards,

Claudio Mulatti, Ph.D.

Academic Editor

PLOS ONE

Additional Editor Comments (optional):

Reviewers' comments:

Reviewer's Responses to Questions

**Comments to the Author**

1. If the authors have adequately addressed your comments raised in a previous round of review and you feel that this manuscript is now acceptable for publication, you may indicate that here to bypass the “Comments to the Author” section, enter your conflict of interest statement in the “Confidential to Editor” section, and submit your "Accept" recommendation.

Reviewer #1: All comments have been addressed

Reviewer #2: All comments have been addressed

2. Is the manuscript technically sound, and do the data support the conclusions?

Reviewer #1: Yes

Reviewer #2: Yes

3. Has the statistical analysis been performed appropriately and rigorously? 

Reviewer #1: Yes

Reviewer #2: Yes

4. Have the authors made all data underlying the findings in their manuscript fully available?

Reviewer #1: Yes

Reviewer #2: Yes

5. Is the manuscript presented in an intelligible fashion and written in standard English?

Reviewer #1: Yes

Reviewer #2: Yes

6. Review Comments to the Author

Reviewer #1: (No Response)

Reviewer #2: The authors have done a very good job of revising the manuscript. They have addressed all of the issues raised in the initial review.

7. PLOS authors have the option to publish the peer review history of their article (what does this mean?). If published, this will include your full peer review and any attached files.

Reviewer #1: Yes: Kenneth R Paap

Reviewer #2: No

---

## [Editor Report · Acceptance letter]

3 Apr 2020

PONE-D-19-34005R1 

A role for the cerebellum in the control of verbal interference: comparison of bilingual and monolingual adults 

Dear Dr. Filippi:

I am pleased to inform you that your manuscript has been deemed suitable for publication in PLOS ONE. Congratulations! Your manuscript is now with our production department. 

With kind regards,

on behalf of

Dr. Claudio Mulatti 

Academic Editor

PLOS ONE